# Diagnostic Approach for Venous Thromboembolism in Cancer Patients

**DOI:** 10.3390/cancers15113031

**Published:** 2023-06-02

**Authors:** Hélène Helfer, Yara Skaff, Florent Happe, Sadji Djennaoui, Jean Chidiac, Géraldine Poénou, Marc Righini, Isabelle Mahé

**Affiliations:** 1Service de Médecine Interne, Hôpital Louis Mourier, Assistance Publique des Hôpitaux de Paris (AP-HP), 92700 Colombes, France; helene.helfer@aphp.fr (H.H.); yara.skaff@aphp.fr (Y.S.); florent.happe@aphp.fr (F.H.); sadji.djennaoui@aphp.fr (S.D.); jean.chidiac@aphp.fr (J.C.); geraldine.poenou@chu-st-etienne.fr (G.P.); 2Université Paris Cité, 75006 Paris, France; 3INSERM UMR-S-1140, 75006 Paris, France; 4FCRIN INNOVTE, 42055 Saint-Étienne CEDEX 2, France; marc.righini@hcuge.ch; 5Service d’Angiologie et Hémostase, HUG—Hôpitaux Universitaires de Genève, 1205 Genève, Switzerland

**Keywords:** cancer, venous thromboembolism, diagnosis, algorithm

## Abstract

**Simple Summary:**

Cancer patients have an increased risk of venous thromboembolic diseases, which are a major cause of morbidity and mortality in this population. The current recommended diagnostic approach consists of a systematic algorithm based on clinical probability, D-dimer measurement, and/or diagnostic imaging. Broad symptoms and elevated D-dimer levels make this diagnosis challenging in cancer patients. To improve venous thromboembolism exclusion, several approaches have been developed, such as ordering systematic imaging tests, new diagnostic algorithms based on clinical probability assessment, and adjusted D-dimer thresholds. However, there is still a lack of dedicated diagnostic algorithm specific for this population.

**Abstract:**

Venous thromboembolic disease (VTE) is a common complication in cancer patients. The currently recommended VTE diagnostic approach involves a step-by-step algorithm, which is based on the assessment of clinical probability, D-dimer measurement, and/or diagnostic imaging. While this diagnostic strategy is well validated and efficient in the noncancer population, its use in cancer patients is less satisfactory. Cancer patients often present nonspecific VTE symptoms resulting in less discriminatory power of the proposed clinical prediction rules. Furthermore, D-dimer levels are often increased because of a hypercoagulable state associated with the tumor process. Consequently, the vast majority of patients require imaging tests. In order to improve VTE exclusion in cancer patients, several approaches have been developed. The first approach consists of ordering imaging tests to all patients, despite overexposing a population known to have mostly multiple comorbidities to radiations and contrast products. The second approach consists of new diagnostic algorithms based on clinical probability assessment with different D-dimer thresholds, e.g., the YEARS algorithm, which shows promise in improving the diagnosis of PE in cancer patients. The third approach uses an adjusted D-dimer threshold, to age, pretest probability, clinical criteria, or other criteria. These different diagnostic strategies have not been compared head-to-head. In conclusion, despite having several proposed diagnostic approaches to diagnose VTE in cancer patients, we still lack a dedicated diagnostic algorithm specific for this population.

## 1. Introduction

Venous thromboembolism (VTE) including pulmonary embolism (PE) and deep vein thrombosis (DVT) has a well-known association with cancer [1]. The risk of VTE in cancer patients, also called cancer-associated thrombosis (CAT), is about seven times higher compared to the general population [2]. Given the fact that the occurrence of a VTE is life threatening and that effective treatments are available, cancer patients are made aware of VTE signs and symptoms. Therefore, the suspicion of VTE is a frequent reason for emergency department visits.

The diagnostic approach for suspected VTE is well established in the general population [3], whereas in cancer patients a lower efficiency has been reported [4]. For any patients with suspected PE or DVT signs, the diagnostic strategy to exclude PE or DVT starts with a clinical probability. In patients with high clinical probability, PE or DVT likely, imaging should be requested: CT pulmonary angiogram (CTPA) or ventilation/perfusion (V/Q) scanning in patients with chronic kidney disease (GFR < 30 mL/min) or contrast/iodine allergy vs. ultrasound in patients with PE or DVT suspicion, respectively. In patients with low or intermediate clinical probability or PE or DVT unlikely, D-dimer is recommended. If D-dimer returns a negative result then PE or DVT are excluded. If D-dimer is positive, imaging should be ordered as previously mentioned [4,5]. The elevation of the biomarker D-dimer is an important predictive tool that also has a high negative predictive value (NPV), in both cancer and noncancer patients, if it is used in association with a low or an intermediate clinical probability or an unlikely clinical probability [6]. However, D-dimer tends to be more frequently elevated in patients with cancer, reducing its clinical usefulness [7]. Yet, there are no dedicated guidelines for a diagnostic approach to VTE in cancer patients. This review article discusses the diagnostic approach of VTE in cancer patients.

## 2. Clinical Probability of Venous Thromboembolism Events in Cancer Patients

The combination of predisposing factors, symptoms and clinical findings allows the classification of patients with suspected VTE into different categories of clinical probability of VTE, which correspond to a higher actual prevalence of confirmed PE. This pre-test assessment can be done empirically or by using prediction rules which are a key step in the diagnostic algorithms of VTE.

When using Wells or Geneva scores, the most commonly used prediction rules, the presence of cancer on its own increases directly the clinical probability of VTE. In fact, the revised Geneva score includes 8 variables from which having an active cancer already contributes to two points for a threshold of 4 points to be classified in the moderate probability category [8]. The DVT Wells score includes 9 variables from which having an active cancer accounts for 1 point leading to the moderate probability category [9]. Consequently, as soon as a cancer patient presents a suggestive symptom, he is often classified as having a moderate or a high clinical probability [8,9].

In their study, Douma et al. evaluated the ability of Wells score to diagnose PE in both cancer and non-cancer patients and confirmed that its competence was lower in cancer patients with 18% of them having low probability compared to 69% in non-cancer patients [10]. In order to evaluate the diagnostic value of Wells’ score parameters, a multivariable logistic regression model was used to study the odds ratios for all the parameters. The only discriminative parameter for PE in cancer population was “other diagnosis less likely than PE” (OR, 3.7; 95% CI, 2.2–6.3) [10]. Furthermore, two studies evaluating Wells criteria for DVT found that cancer patients were not likely to be categorized in the low risk group. Consequently, between cancer and non-cancer patients, less than 20% versus 58% to 62.9% were in the low category or unlikely probability respectively [11,12].

## 3. D-Dimer in Cancer Patients with Suspected VTE

The levels of D-dimer are often increased nonspecifically in cancer patients [10]. In fact, a hypercoagulable state exists frequently and is related to the high fibrinogen, von Willbrand factor, and secondarily FVIII whenever an inflammatory state is present. This results in an elevation of the thrombin production that is combined with the resistance of the activated protein C leading to the production of excess fibrin and thus to a reactive fibrinolysis. Furthermore, some malignant cells are directly involved in the activation process of the coagulation and/or fibrinolysis. D-dimer levels are usually above the exclusion threshold for VTE, even when adapted to age, leading to the use of imaging in comorbid and sick patients [13,14,15]. This was confirmed in a multicenter management study that gathered inpatients and outpatients with suspected PE from 12 hospitals in the Netherlands. Out of 3306 patients with PE suspicion, 475 (14%) had active malignancy at presentation. All patients had an assessment of clinical probability, a measurement of D-dimer, and an imaging. In cancer patients, 51% were found to have a low clinical probability with only 8.4% (40 patients) having a D-dimer level below the cutoff value of 500 ng/mL and 12% (57 patients) having D-dimer levels below the age-adjusted D-dimer cutoff value. At the end, the diagnosis of PE was excluded in 10% of cancer patients based on the clinical probability calculation and the measurement of D-dimers.

In addition, data regarding the performance (sensitivity, specificity, NPV) and added value of D-dimer measurement in clinical practice in cancer patients are limited. Most of the published studies are retrospective or subgroup analyses and include a small number of cancer patients. Nevertheless, studies based on D-dimer measurement report a high NPV in cancer patients, varying between 94% to 100% for ELISA/ELFA tests. Less good NPV have been reported for latex agglutination tests [16,17] (Table 1). As already mentioned, the specificity of D-dimer in DVT and PE diagnosis is low in cancer patients, resulting in a low proportion of patients in whom VTE can be ruled out [17].

An analysis of 3 different prospective studies consisting of 2496 patients with suspected DVT including 200 cancer patients showed that in cancer population a low or moderated clinical probability score was correlated with high sensitivity and NPV (low probability = 99.6% 95% CI: 98.6–99.9), intermediate probability = 99.3% CI 95 (98.5–99.7)) and without cancer (low probability 95% CI throughout the manuscript = 100% CI 95 (69.8–100), intermediate probability = 100% CI 95 (82.8–99.6)) [11]. The NPV in cancer patients was 100% but with a large confidence interval (CI) going up to 69.8% because of a small sample number [11]. Additionally, Di Nisio et al. endorsed these results in their retrospective study in patients with suspected DVT (Table 1) [21]. Those studies demonstrated that only 6 to 9% of cancer patients presented a low clinical probability score and a low D-dimer level, allowing the exclusion of the VTE diagnosis without imaging. In other words, more than 90% of cancer patients still need imaging to exclude VTE in addition to the biological test [11,21]. Plus, when DVT is suspected, if the 2-level Wells score is used, the proportion of cancer patients for whom no imaging examination is required increases to 12%; however, this still means that 88% of cancer patients need imaging [11].

In a meta-analysis by Van Es et al. [22], including individual patient data from 6 prospective studies encompassing a total of 7268 patients with 938 patients having an active cancer, 85 patients with cancer (9.1%) had D-dimer levels below the threshold of 500 ng/mL, whereas in patients without cancer, 1904 patients (30.4%) had a D-dimer level below 500 ng/mL [22]. These results emphasize the contrast of the efficacy and safety of the combined strategy of using the clinical probability with D-dimer between cancer and noncancer population. Approximately, one-third of the patients without cancer may have the diagnosis of VTE ruled out by only using the combined strategy with a high NPV, which is not the case in cancer patients [9,23]. Moreover, clinicians perceive that D-dimer concentrations are mostly increased in cancer patients so they tend to proceed directly with imaging to rule out or diagnose VTE [24]. More recently, studies have been conducted to adjust the D-dimer threshold according to the age of cancer patient or to the clinical probability like in YEARS and PEGeD [16,25] (Figure 1).

## 4. D-dimer Adjusted to Age in Cancer Patients with Suspected VTE

The age-adjusted D-dimer cutoff strategy was applied in patients older than 50 years old and consists of multiplying age by 10 (age × 10 µg/L) in contrast with the standard cut-off value of 500 µg/L. The use of this new approach has been tested in cancer patients with suspected PE to improve D-dimer specificity without decreasing sensitivity. Combined data from two prospective multicenter studies of patients with suspected PE showed that doubling D-dimer cutoff from 500 to 1000 ng/L enhanced specificity from 16 to 37% in cancer patients, still keeping an elevated NPV at 98% (95% CI, 85–99%) [15]. In fact, after increasing D-dimer threshold to 1000 ng/mL, the exclusion of PE increased from 11 to 26% in cancer patients [15]. However, the large CI for NPV (95% CI, 85–99%) caused by the limited number of patients led to a cautious interpretation of these results. In addition, a post hoc subgroup of the ADJUST-PE study assessing the diagnostic performance of an age-adjusted D-dimer threshold in outpatients with suspected PE found that the proportion of cancer patients with a D-dimer concentration below the traditional threshold of 500 ng/mL was 9.9% compared to 19.7% using the age-adjusted cutoff [26]. Despite the twofold increase in the percentage of patients with cancer having D-dimer levels under the threshold, it is inferior to noncancer patients (19.7% vs. 41.9%; *p* < 0.001) [26,27]. Future data are needed to confirm that a higher PE diagnosis exclusion rate in a larger number of cancer patients is associated with a higher D-dimer threshold. In his meta-analysis, Van Es et al. studied D-dimer performance in both its classic and age-adjusted form [22]. Using the age-adjusted D-dimer cutoff value increased the efficiency of D-dimer testing from 28% to 33% but kept a NPV lower than the acceptable safe threshold [22]. The results of these studies suggest that the combination of a low probability on Wells score plus a negative D-dimer level are not enough to rule out VTE in patients with cancer. The age-adjusted D-dimer cutoff could improve the efficacy of the algorithm yet 77% of cancer patients would require imaging to rule out VTE diagnosis.

## 5. New Algorithms Based on D-dimer Adjusted to Clinical Probability in Cancer Patients with VTE Suspicion

The PEGeD study evaluated the diagnostic approach of PE exclusion using Wells score with D-dimer cutoffs adapted to clinical pretest probability (CPTP) [16]. Patients were divided into three categories: low, moderate, and high CPTP. PE was excluded without imaging in patients with low CPTP and D-dimer levels less than 1000 ng/mL or patients with moderate CPTP and D-dimer less than 500 ng/mL. The outcomes of this study were comparable with prior studies about the efficiency of D-dimer adjusted to clinical probability as the number of CT pulmonary angiograms (CTPAs) was reduced by 17.6% in a noncancer population (when compared to the standard threshold of 500 ng/mL) without false negative (0% IC 95% 0.0–0.29). Unfortunately, this study did not analyze a subgroup of cancer patients [16].

YEARS is a new algorithm that was proposed to also lower the number of CTPAs exams in every age cohort [25]. The YEARS study consisted of 3465 patients with suspected PE that were taken in charge by evaluating the YEARS clinical criteria containing three items (clinical feature of DVT, hemoptysis, and PE as the most likely diagnosis) and the D-dimer levels. In patients that did not have any YEARS items and a D-dimer level under 1000 ng/mL or in patients with at least one positive item form YEARS and a D-dimer level under 500 ng/mL, PE was considered as directly excluded whereas all remaining patients had imaging. This study showed that, in an additional 14% and 7.6% of patients, we can safely rule out PE without imaging compared to the Wells score with fixed D-dimer threshold and age-adjusted threshold, respectively. However, this method had a higher false negative rate (2.6%, IC 95% 1.3–5.2) in the restricted subgroup of cancer patients (336 out of 3465 included) against 0.41% (IC 95% 0.23–0.73) in noncancer patients followed up for 3 months. To note, the absolute false negative number was small in the subgroup of cancer patients, where 7 patients still had PE out of the 279 patients that had PE excluded at the beginning, but the majority of patients had additional risk factors of VTE, mainly surgery [25]. The same rate of false negative was seen in a meta-analysis applying the Wells score and a fixed D-dimer threshold <500 ng/mL (2.6%, CI 95% 0.57–11.0) [22].

## 6. Discussion

Venous thromboembolic disease is a frequent, life-threatening, and treatable disease. Early diagnosis being essential, different diagnostic strategies are available to improve the identification of patients at risk and limit the use of imaging to rule out VTE. The first step in the actual recommended diagnostic algorithm starts with a clinical probability score. When the clinical probability is high, imaging is directly indicated, whereas in case of low or moderate probability or unlikely clinical probability, D-dimer level should be assessed. In the case that D-dimers are below the threshold, VTE can be ruled out.

In the case of VTE suspicion, cancer patients most frequently have a moderate or high-risk clinical probability. This is explained by the presence of “cancer” as one of the criteria included in all probability scores, in addition to the high incidence of VTE symptoms (for example: dyspnea or chest pain) in these patients. Furthermore, some malignant cells are directly involved in the activation process of the coagulation and or fibrinolysis. D-dimer levels were noted to elevated in cancer patients [28], which led them to be above the exclusion threshold for VTE, even when adapted to age, leading to the use of imaging in comorbid and sick patients. This was demonstrated in a large study where 88% to 94% of patients with cancer needed additional investigations into D-dimer measurement; making D-dimer alone less helpful to rule out PE in cancer patients [11].

Several alternatives to the measurement of D-dimer were researched, especially after assessing the diagnostic value of an inflammatory biomarker such as CRP alone or associated with a clinical probability score in patients with suspected PE against the D-dimer measurement level. It was concluded that the measurement of CRP alone or associated with a clinical probability cannot safely exclude the diagnosis of PE [29]. For all these reasons, various strategies were used to improve the performance of the diagnostic algorithms in cancer patients.

The first strategy consisted of conducting imaging on all patients, which overexposed them to contrasting products and radiations even though they are often known to have several co-morbidities and renal insufficiency [30].

The second approach was to develop new algorithms that take into consideration the clinical probability with different D-dimer thresholds such as, for example, the YEARS algorithm, which shows promise in improving the diagnosis and management of PE in cancer patients. A prospective randomized study called HYDRA, which is currently in progress (Netherlands Trial Register NL7752), studies the safety and efficacy of the YEARS algorithm compared to CTPA alone in cancer patients. The results are expected by the end of 2024 and will contribute to orienting the optimal diagnostic approach in daily practice and especially in this high-risk population. While waiting, an external validation of the YEARS algorithm was carried out in an independent cohort. This one confirmed that the algorithm safely excludes more PE diagnosis in the cohort [31]. Meanwhile, the current guidelines do not provide clear recommendations on the diagnostic approach for VTE suspicion in cancer patients [3,4].

The third approach used an adjusted D-dimer threshold, to age, pretest probability, clinical criteria, or YEARS criteria, seems safe in cancer patients allowing the increase in the clinical usefulness of the association between the clinical probability score and D-dimer, which restricts the overusing of imaging [32,33]. Stals et al., in their study about the “safety and efficiency of diagnostic strategies for ruling out PE in clinically relevant subgroups”, found that predicted failure rates were greatest with an adapted D-dimer threshold. Failure rates fluctuated between 2% and 4% in the different patient subgroups [34]. In addition, this strategy would be translated by a decrease in the detection of subsegmental PE, mainly non-pertinent, as well as lower costs [11].

## 7. Perspective and Limitations

All the strategies previously cited were validated in patients presenting to the emergency department and none were validated in patients that were hospitalized. It will be interesting to study the performance of a new combination of tests in hospitalized patients with a suspicion of VTE. Moreover, in real life, applying these new algorithms may be very challenging due to various obstacles such as education, training, technical issues, and cost-effectiveness. Exploring these methods’ implementation and potential challenges would be an important topic in future research to provide practical guidance. Last but not least, comparing the effectiveness of the existing diagnostic strategies in different patient subgroups such as tumor types, stages of the disease, and treatment plan would be an important topic for improving the diagnosis of VTE in cancer patients.

## 8. Conclusions

The current diagnostic algorithm of VTE is not reliable in cancer patients since combining the clinical probability score and the D-dimer level, as suggested, is less efficient in this population. Despite having several proposed strategies to limit the use of imaging, we still lack a cautious and dedicated diagnostic approach to VTE in this population. Hence, there is a need for further studies to develop a systematic, efficient, and accurate diagnostic algorithm of VTE in cancer patients.

## Figures and Tables

**Figure 1 cancers-15-03031-f001:**
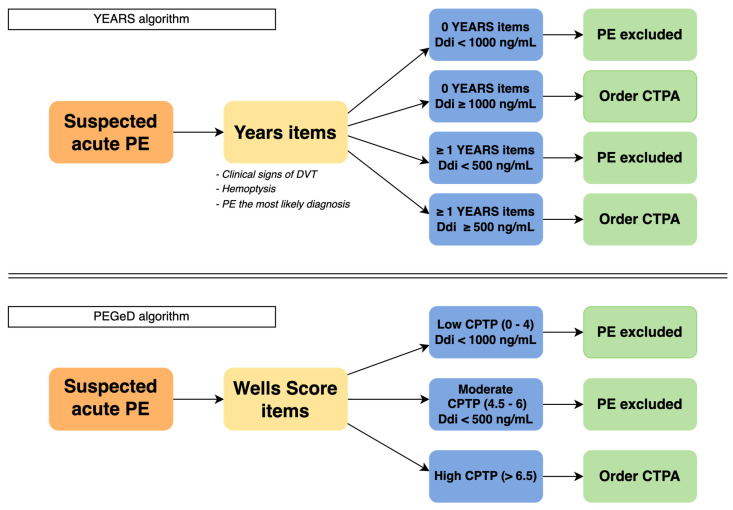
YEARS and PEGeD algorithms with a D-dimer threshold adapted to clinical items. PE, pulmonary embolism; CPTP: clinical pretest probability; CTPA: computed tomography pulmonary angiography [16,25].

**Table 1 cancers-15-03031-t001:** Characteristics of the major cohort studies with a suspicion of VTE and D-dimer adjusted to age [17]. Technique of D-dimer Measurement: ELFA Vidas^®^ (rapid ELISA, bioMérieux, France), agglutination latex beads (STA-Lia test or Tinaquant, Roche, Germany).

Reference	Types of VTE	Number of Patients (% Man)	Mean Age in Years (SD)	Prevalence of VTE (%)	Patients Studied	Technique of D-dimer Measurement	Probability of Clinical Score Used	% of Patients with VTE Excluding Cancer Subgroup
**Douma et al., 2010 [10]**	EP	1721 (41)	61 (19)	24	Outpatients presetting to ER or clinics	ELFA (Vidas^®^)	Wells score (≤4)	10
**Douma et al., 2010 [10]** **Validation cohort**	EP	1819 (49)	59 (19)	21	Outpatients presetting to ER or clinics	ELFA (Vidas^®^)	Revised Geneva score	7
**Penaloza et al., 2012 [18]** **French cohort**	EP	1529 (39)	ND	28	Outpatients presetting to ER or clinics	ELFA (Vidas^®^) or agglutination latex beads	Revised Geneva score	7.6
**Penaloza et al., 2012, [18]** **European cohort**	EP	1645 (42)	59	18	Outpatients presetting to ER or clinics	ELFA (Vidas^®^) or agglutination latex beads	Revised Geneva score	7.6
**Penaloza et al., 2012 [18]** **American cohort**	EP	7940 (33)	49	51	Outpatients presetting to ER or clinics	ELFA (Vidas^®^) or agglutination latex beads	Revised Geneva score	6.2
**Douma et al., 2010 [10] ** **Validation cohort 1**	EP	3306 (43)	53 (18)	20	Outpatients or inpatients	ELFA (Vidas^®^) or agglutination latex beads	Wells score	14
**Van Es, 2012 [19]**	EP	456 (46)	65	27	Outpatients or inpatients	ELFA (Vidas^®^) or agglutination latex beads	Wells score	10
**Schouten, 2013 [17]**	DVT	1374 (27)	59 (17)	20	Patients in ED	ELFA (Vidas^®^) or agglutination latex beads	Wells score	9
**Douma, 2012 [20] cohort 1**	DVT	812 (36)	59 (17)	39	Outpatients presetting to ER or clinics	Agglutination latex beads	Wells score	NR
**Douma, 2012 [20] cohort 2**	DVT	474 (38)	61 (19)	23	Outpatients presetting to ER or clinics	ELFA (Vidas^®^)	Clinical score evaluated by the physician	NR
**Douma, 2012 [20] cohort 3**	DVT	359 (41)	66 (17)	23	Outpatients presetting to ER or clinics	Agglutination latex beads	Score de Wells	NR
**Douma, 2012 [20] cohort 4**	DVT	536 (38)	65 (16)	10	Outpatients presetting to ER or clinics	Agglutination latex beads	Score de Wells	NR
**Douma, 2012 [20] cohort 5**	DVT	617 (52)	58 (18)	37	Outpatients presetting to ER or clinics	Agglutination latex beads	Score de Wells	NR

## Data Availability

The data presented in this study are available in this article.

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
