# Peer review of "Diagnostic Approach for Venous Thromboembolism in Cancer Patients"

_cancers, 2023, doi:10.3390/cancers15113031_

Round 1

Reviewer 1 Report

The review is well written and exhaustive

The topic is of great interest  for oncologists

Conclusions are disappointing but inevitable

Author Response

Thank you so much for your feedback. 

Reviewer 2 Report

The authors discussed about Venous thromboembolic disease (VTE) in cancer patients.

This is a very nice and well written literature revision.

I have only few comments.

1) The authors should discuss cardiac thrombi mainly in the right chambers, because it is a frequently remark in patients with cancers (please cite: DOI: 10.1016/j.jcmg.2022.06.005 )

2) The authors did not discuss the role of non-invasive diagnostics tool 

Minor english revision

Author Response

  • The authors should discuss cardiac thrombi mainly in the right chambers, because it is a frequently remark in patients with cancers (please cite: DOI: 10.1016/j.jcmg.2022.06.005 )

Thank you so much for your feedback. Indeed, cardiac thrombi are remarked in cancer patients but it was not in the objectives of our review to discuss the types of VTE in this population. We focused on the diagnostic approach to diagnose VTE. Yet, it is an important topic and it deserves to be treated in a separate review.  

  • The authors did not discuss the role of non-invasive diagnostics tool 

Thank you for pointing this out. Despite exploring in depth clinical probability and d-dimer (that cause issue compared to general population) in the diagnostic algorithm to diagnose or exclude VTE in cancer patients, we completely agree with your feedback. As per this assessment, we discussed non-invasive tools (CT Pulmonary Angiogram, V/Q scan and ultrasound) in our introduction.

Reviewer 3 Report

This article mainly explores the diagnostic methods and strategies for venous thromboembolism (VTE) in cancer patients. The article first introduces the high prevalence of VTE in cancer patients and the importance of diagnosis and analyzes the current VTE diagnostic methods and their limitations. The article delves into a series of new diagnostic strategies, such as a new algorithm based on age and clinical probability adjusted by D-dimer levels, such as the YEARS algorithm. The article also compares and evaluates different experimental methods to determine the optimal diagnostic strategy. In addition, the article discusses future research directions and challenges, and the conclusion states that although various strategies aim to limit imaging examinations, the diagnostic methods for VTE in cancer patients still need improvement. I believe the article has room for improvement, and I hope to discuss the following points with the author. If the author feels my suggestions are reasonable, appropriate modifications can be made to the article.

 1.         In the background section regarding the YEARS algorithm, please provide more development background and applicable situations to help readers understand its potential value.

 2.         In the background section, the author mentioned the elevated D-dimer levels in cancer patients but did not discuss the specific reasons in detail. Is this related to tumor type, disease course, and treatment methods?

 3.         In the results section, the author mentions the D-dimer threshold adjustment strategy but does not provide details on determining and adjusting the appropriate threshold in clinical practice. The author may consider discussing threshold determination methods, individualized threshold adjustments, threshold adjustment frequency, and indicators.

 4.         When evaluating the effectiveness of existing diagnostic strategies, the author mainly focuses on the overall cancer patient population but does not explicitly mention comparisons of differences between different patient subgroups (such as different tumor types, stages of the disease, and treatment plans). Future research can consider a more in-depth analysis of these different patient subgroups to understand better the applicability and effectiveness of existing diagnostic strategies in each subgroup.

 5.         In the discussion section, the article mentions new algorithms and diagnostic methods but does not discuss their application and challenges in clinical practice in detail. In the real world, applying these algorithms may face various challenges, such as education and training, technical issues, interdisciplinary collaboration, guideline policies, and cost-effectiveness. The author can further explore these methods' implementation and potential challenges in clinical practice in future research to provide practical guidance.

 6.         In this article, the author mainly focuses on applying D-dimer and CRP in the VTE diagnosis of cancer patients and does not discuss other potential biomarkers and their value in VTE diagnosis, such as plasminogen activator inhibitor-1 (PAI-1), tissue factor pathway inhibitor (TFPI), etc. Future research can consider exploring the potential application of other biomarkers in the VTE diagnosis of cancer patients to optimize existing diagnostic methods.

 7.         In the results section, the author could add bar charts to assist in the analysis. This will help improve the readability and attractiveness of the article. For example, a comparison of the VTE detection rate using different diagnostic methods: this bar chart can display the impact of different diagnostic methods, such as D-dimer, CRP, and new algorithms, on the VTE detection rate. This will help to intuitively compare the pros and cons of various diagnostic methods and provide guidance for clinical practice.

 8.         In the discussion section, the author may consider discussing the role and limitations of other potential diagnostic tools in the VTE diagnosis of cancer patients. For example, radionuclide lung ventilation/perfusion scan (V/Q scan), MRI, ultrasound, magnetic resonance venography (MRV), etc. Also, analyze the application prospects of other non-invasive diagnostic techniques (such as bioimpedance technology, hemodynamic analysis, etc.) in the VTE diagnosis of cancer patients. In future research, the author may consider discussing the application of these diagnostic tools and comparing their advantages and disadvantages in cancer patients. Combining multiple diagnostic methods may be more beneficial for improving diagnostic accuracy.

 9.         In the conclusion section, to improve foresight and practicality, the author may consider mentioning new biomarker identification and validation, individualized diagnostic strategy development, interdisciplinary collaboration, new technology and diagnostic tool evaluation, and VTE prevention strategies in future work. It will provide readers with a broader perspective and practical advice, promoting the development and innovation of VTE diagnosis in cancer patients.

Author Response

  1. In the background section regarding the YEARS algorithm, please provide more development background and applicable situations to help readers understand its potential value.

Thank you so much for this feedback : we changed the organization of the paragraphs to highlight the background and potential value of the new algorithm such as YEARS. 

  1. In the background section, the author mentioned the elevated D-dimer levels in cancer patients but did not discuss the specific reasons in detail. Is this related to tumor type, disease course, and treatment methods?

That’s actually a great observation and thank you for pointing this out. Elevated D-dimer levels are often found in a number of clinical settings and in patients without proven venous thrombosis such as in malignancy, pregnancy, infection, arrhythmia, trauma… and clinicians may find such results difficult to explain. Great point and as per this feedback we did add an explanation of the elevation of D-dimer in cancer patients.

  1. In the results section, the author mentions the D-dimer threshold adjustment strategy but does not provide details on determining and adjusting the appropriate threshold in clinical practice. The author may consider discussing threshold determination methods, individualized threshold adjustments, threshold adjustment frequency, and indicators.

Thank you so much for this feedback. We completely agree, and as per this feedback, we added more details on threshold adjustment.   

  1. 4.         When evaluating the effectiveness of existing diagnostic strategies, the author mainly focuses on the overall cancer patient population but does not explicitly mention comparisons of differences between different patient subgroups (such as different tumor types, stages of the disease, and treatment plans). Future research can consider a more in-depth analysis of these different patient subgroups to understand better the applicability and effectiveness of existing diagnostic strategies in each subgroup.

This is a great point, thank you. Indeed, future research should consider comparing different categories in order to improve the use of the existing strategies in each subgroup. This point was added.

  1. In the discussion section, the article mentions new algorithms and diagnostic methods but does not discuss their application and challenges in clinical practice in detail. In the real world, applying these algorithms may face various challenges, such as education and training, technical issues, interdisciplinary collaboration, guideline policies, and cost-effectiveness. The author can further explore these methods' implementation and potential challenges in clinical practice in future research to provide practical guidance.

Another great point, thank you. Very important for future research, as per this feedback the above point was added. 

  1. In this article, the author mainly focuses on applying D-dimer and CRP in the VTE diagnosis of cancer patients and does not discuss other potential biomarkers and their value in VTE diagnosis, such as plasminogen activator inhibitor-1 (PAI-1), tissue factor pathway inhibitor (TFPI), etc. Future research can consider exploring the potential application of other biomarkers in the VTE diagnosis of cancer patients to optimize existing diagnostic methods.

Thank you for this feedback. The other potential biomarkers mentioned above are not used in routine patient care. They are still not validated and thus not available to be used in daily practice. We discussed approved and available parameters

  1. In the results section, the author could add bar charts to assist in the analysis. This will help improve the readability and attractiveness of the article. For example, a comparison of the VTE detection rate using different diagnostic methods: this bar chart can display the impact of different diagnostic methods, such as D-dimer, CRP, and new algorithms, on the VTE detection rate. This will help to intuitively compare the pros and cons of various diagnostic methods and provide guidance for clinical practice.

Thank you for this feedback. Unfortunately, there is no enough available data in this field. The studies are manly done in general population and the subgroup of cancer patients had a small sample.

  1. In the discussion section, the author may consider discussing the role and limitations of other potential diagnostic tools in the VTE diagnosis of cancer patients. For example, radionuclide lung ventilation/perfusion scan (V/Q scan), MRI, ultrasound, magnetic resonance venography (MRV), etc. Also, analyze the application prospects of other non-invasive diagnostic techniques (such as bioimpedance technology, hemodynamic analysis, etc.) in the VTE diagnosis of cancer patients. In future research, the author may consider discussing the application of these diagnostic tools and comparing their advantages and disadvantages in cancer patients. Combining multiple diagnostic methods may be more beneficial for improving diagnostic accuracy.

Thank you for bringing this up. Despite exploring in depth clinical probability and d-dimer (that cause issue compared to general population) in the diagnostic algorithm to diagnose or exclude VTE in cancer patients, we completely agree with your feedback. As per this assessment, we discussed non-invasive tools (CT Pulmonary Angiogram, V/Q scan and ultrasound) in our introduction.

  1. In the conclusion section, to improve foresight and practicality, the author may consider mentioning new biomarker identification and validation, individualized diagnostic strategy development, interdisciplinary collaboration, new technology and diagnostic tool evaluation, and VTE prevention strategies in future work. It will provide readers with a broader perspective and practical advice, promoting the development and innovation of VTE diagnosis in cancer patients.

Thank you for this feedback. As per the feedback, we dedicated a new paragraph to discuss perspective and limitations and provide broader perspective to the readers.